# Prevalence of Infectious Agents Causing Abortion in Pregnant Women Using Serological Tests and Histopathological Analysis

Ahmed M. Mahmoud [1,*], Howaida Mahmoud Hagag [2,3], Khadiga Ahmed Ismail [2], Abeer Muslih Alharthi [2], Amal Amer Altalhi [2], Najwa F. Jaafer [4], Hassna H. Alharthi [4], Ahmed A. Elwethenani [5], Khadiga H. Khan [5], Seham Hazza Al-ajmani [5], Alaa Khader Altalhi [5], Abdullah S. Al-Ghamdi [5], Naïf Saad Althobaiti [5], Reem Amr Ramadan [6] and Osama M. Khalifa [7]

1    Department of Urology, Mayo Clinic, Rochester, MN 55905, USA
2    Department of Clinical Laboratory Sciences, College of Applied Medical Sciences, Taif University, Taif 21974, Saudi Arabia; howaida1210@yahoo.com (H.M.H.); khadigaah.aa@tu.edu.sa (K.A.I.)
3    Department of Pathology, Faculty of Medicine, Al-Azhar University, Cairo 11884, Egypt
4    Gynacology and Obstetrics Department, King Faisal Medical Complex, Taif 26724, Saudi Arabia; dr.hsna@yahoo.com (H.H.A.)
5    Laboratory Department, King Faisal Medical Complex, Taif 26724, Saudi Arabia
6    Department of Internal Medicine, MTI Faculty of Medicine, Cairo 11728, Egypt
7    Department of Internal Medicine, Faculty of Medicine, Ain Shams University, Cairo 11566, Egypt; osama_khalifa9955@yahoo.com
\*    Correspondence: mahmoud.ahmed@mayo.edu

**Abstract:** Background: Abortion is a spontaneous loss of pregnancy before 20 weeks. Approximately 42 million pregnancies end in abortion. The maternal infections that are transmissible from mother to fetus are caused by many pathogens, of which the TORCH complex contributes majorly to neonatal and infant deaths globally. The aim of this study is to detect the prevalence and types of infectious causes of abortion. One hundred aborted women admitted to King Faisal Medical Complex Maternity Hospital in Taif City between the period of 2018 and 2020 were enrolled in this study. The serological test reports (TORCH panel), as well as reports of hematological (CBC) and chemical parameters, were obtained from laboratory management system databases, reviewed, and then analyzed. The H&E-stained microscopic slides of their product of conception (POC) were examined under a microscope and compared with histopathological reports. The prevalence of TORCH infections was 8% in aborted women. *Hepatitis B virus* (HBV) and mixed TORCH infections constituted the highest percentage of TORCH pathogens in aborted women, constituting 6%. The most detected histopathological finding in seropositive cases (50%) was POC, with mixed inflammatory infiltrates and chronic endometritis, while in seronegative aborted women, POC was normal (64.1%). There is a statistically significant increase in the mean count of white blood cells in seropositive women. Therefore, it is important to provide health campaigns to bring awareness to the population about the risk factors of infectious agents to be avoided, especially during pregnancy.

**Keywords:** abortion; TORCH infection; histopathology

## 1. Introduction

A spontaneous pregnancy loss before 20 weeks is referred to as an abortion (miscarriage), which used to be referred to as a spontaneous abortion (SA) [1]. Early miscarriage is the term for pregnancy loss that occurs in as many as one in five pregnancies during the first trimester (less than 12 weeks of gestation). One to two percent of pregnancies end in a late miscarriage during the second trimester (12 to 24 weeks of gestation). Evidence suggests that easily treatable infections may cause up to 15% of miscarriages that occur early and up to 66% of miscarriages that occur late [2].

Threatened miscarriage, inevitable miscarriage, incomplete miscarriage, and full miscarriage are the stages of spontaneous miscarriage, according to the World Health

Organization (WHO) [1]. The term "recurrent pregnancy loss" (RPL), sometimes known as "recurrent miscarriage" or "habitual abortion," originally refers to three successive pregnancies lost prior to 20 weeks following the last menstrual period. Based on the frequency of spontaneous pregnancy loss, recurrent pregnancy loss occurred about once per 300 pregnancies. However, epidemiological research has found that 1% to 2% of women endure repeated pregnancy loss [3].

Depending on the gestational and maternal ages, the pathophysiology of recurrent pregnancy loss differs, yet many processes may ultimately converge on a single pathway that causes the pregnancy loss. Chromosomal abnormalities, structural uterine abnormalities, and autoimmune diseases are typical mechanisms [4].

The TORCH group, which is an acronym made up of the first letters of the following pathogens' names (*Toxoplasma gondii*, other pathogens, *Rubella virus*, *Cytomegalovirus*, *Herpes simplex virus*), is comprised of the primary pathogens that could cross the placenta after infecting a pregnant woman and cause severe harm to the fetus [5]. The majority of TORCH infections result in minor maternal sickness, while the fetal effects can be severe [6].

The primary TORCH infection during the different stages of pregnancy has serious life-threatening consequences on a fetus in comparison to recurrent infections and may cause spontaneous abortions, congenital malformations, intrauterine growth restriction, stillbirths, prematurity, and chronic postnatal infections [7]. Infection during the first trimester, the time of organogenesis, has more devastating repercussions than infection during the third trimester, which is often when transmission possibilities are maximum [8].

One of the TORCH pathogens, Toxoplasma gondii, is the only one that is not a virus [8]. Apart from immunocompromised patients whose pregnancy may result in abortion, stillbirth, decreased birth weight, or prematurity, recurrent infection in subsequent pregnancies is unusual. Primary infection with Toxoplasma gondii can cause fetal death and miscarriage [9].

A mild, self-limiting viral virus that affects people all throughout the world is Rubella. It is brought on by a member of the Togaviridae family, not carried by an arachnid. Due to the asymptomatic nature of the infection, at least half of all primary Rubella infections go untreated. Congenital Rubella Syndrome (CRS), which affects all organs in the developing fetus, and intrauterine mortality are congenital abnormalities that are linked to maternal Rubella infection during the first trimester [6].

The ubiquitous and species-specific cytomegalovirus (CMV). Humans serve as the virus' reservoir hosts and can spread the infection through direct contact with saliva, urine, and vaginal secretions. Sexual activity or direct contact with contaminated saliva or urine from young children can transmit the disease to pregnant women. The signs and symptoms in newborns include anemia, thrombocytopenia purpura, chorioretinitis, hepatosplenomegaly, microcephaly with cerebral calcification, and intrauterine growth retardation. CMV infection is also a cause of serious childhood impairments, including loss of vision, hearing, and cognitive function [6].

The most prevalent sexually transmitted viral disease in the world is caused by the herpes simplex virus (HSV). HSV1 is spread by nonsexual contact during childhood, but HSV2 is usually sexually transmitted and is the main cause of genital herpes. Primary genital HSV infection is asymptomatic in more than 75% of patients. This infection continues to be a leading cause of morbidity and mortality in neonates. Congenital and neonatal herpes, spontaneous miscarriage, and preterm can all result from genital herpes (HSV) infection during pregnancy [6]. Premature labor, miscarriage, congenital herpes, and newborn herpes can all result from either HSV-1 or HSV-2 infection in an expectant woman [10].

According to a study by Al-Hakami et al. [10], the incidence of TORCH agents is still high in KSA and can cause a variety of congenital illnesses as well as other consequences. Thus, prenatal mothers must undergo routine screening in order to prevent TORCH problems, which cause pregnancy- and neonatal-related morbidity [10].

Since a pregnant woman's immune reactivity determines her risk of contracting a placental or fetal infection, immunological tests should first check her immunoglobulin

levels and look for autoantibodies and then move on to check her T and B lymphocyte subset and natural killer cell (NK) cytotoxic activity [9].

First-trimester specimens have very different compositions, making histopathological evaluation crucial for detecting previously undetected illnesses. The therapy of patients with sporadic and recurrent early pregnancy failure includes histopathology on a regular basis [11].

It is crucial to detect maternal illness early and to monitor the fetus after diagnosis. Knowing about these illnesses can help the physician properly advise mothers on how to avoid becoming infected and help with counseling parents about the possibility of negative fetal outcomes when these infections are present [6].

There is a lack of studies and data about the common types of causative infectious agents as well as other causes of abortion in the Taif population. The present study aimed to determine the common infectious causes and other causes to avoid pregnancy loss as well as help in planning preventive methods. Therefore, the aim of this study was to detect the prevalence of infectious causes of abortion in Taif City and identify the causes of abortion, especially infectious agents in Taif City.

## 2. Materials and Methods

This study was a retrospective study on samples collected from 2018 to 2020. The study was conducted in King Faisal Medical Complex maternity hospital. The study involved 100 women whose fetal remains were collected and examined at the hospital's histology department after a spontaneous miscarriage. The patients were chosen based on inclusion and exclusion standards.

Inclusion criteria: all women who have abortions.

Exclusion criteria: patients who have incomplete data in hospital records.

### 2.1. Study Experiments and Method

Products of conception were sent to a histopathology laboratory in 10% formalin, routinely processed in an automatic tissue processor, and embedded in paraffin wax. Three to five serial sections of four-micron thickness were sectioned using a rotary microtome and then stained with hematoxylin and eosin. In order to confirm or rule out conception, the stained section was microscopically examined for the presence of fetal tissues, trophoblasts, or chorionic villi. Other characteristics that might be seen include irregularities in the chorionic villi, the existence of deciduae without chorionic villi, symptoms of inflammation, and proof of any infectious agents that may be the cause of the condition. Blood samples collected for serological tests to detect pathological agents that cause abortion (TORCH panel) results obtained from laboratory management system databases.

### 2.2. Data Collection

Relevant important clinical data for each case from the hospital files in the histopathology lab and medical records were collected retrospectively using a checklist. The records of the patients, including age, gestational age, complaints before the abortion, and clinical diagnosis (on admission). Among other relevant information, personal history of chronic and infectious diseases, history of contact with pet animals, family history of congenital or hereditary diseases, and complete blood count (CBC) were retrieved from the laboratory database and matched with that in the laboratory request forms which had been received with the specimen and archived in the laboratory to ensure harmony.

### 2.3. Data Analysis

Using SPSS version 22.0 (IBM Corporation, Armonk, NY, USA), Shapiro–Wilk test of normality distribution of the data was used to ascertain whether each variable had a normal distribution. Unpaired t-tests, exact Fisher tests, and chi-square tests were all used for data entry and statistical analysis. All two-sided tests were considered statistically significant if the *p*-value was less than 0.05.

*2.4. Ethical Concerns*

The College of Applied Medical Sciences' research committee approved the study's execution—Taif University—after an explanation of the aim of the study. Further, the study was approved by the ethics committee of the King Faisal Medical Complex maternity hospital (H-02-T-123).

## 3. Results

Our study included a total of 100 aborted women. The percentage of women who had a miscarriage at less than 12 weeks of gestation (early miscarriage) was 65%, which is higher than that of women who miscarried between 12–20 weeks of gestation (late miscarriage) 35%.

The total seropositivity rate of TORCH infections was (8%) in Taif City during the period between 2018 and 2020, as illustrated in Table 1. In early abortion, the seropositivity rate for both *Rubella* and *Hepatitis C* virus (HCV) and mixed TORCH infections was (4%). In late abortion, the seropositivity rate for *Hepatitis B virus* (HBV) was (3%), and mixed TORCH infection was (1%). There was no statistical significance in the Fisher exact test.

**Table 1.** Distribution of histopathological findings in seropositive and seronegative aborted women.

| Histopathological Diagnosis | | Seropositive (8) | Seronegative (92) |
|---|---|---|---|
| Normal POC | number | 0 | 59 |
| | % | 0% | 64.1% |
| Septic abortion | number | 1 | 6 |
| | % | 12.5% | 6.5% |
| POC with mixed inflammatory infiltrates and chronic endometritis | number | 4 | 7 |
| | % | 50% | 7.6% |
| Degenerated POC | number | 2 | 12 |
| | % | 25% | 13% |
| POC with Hydropic changes | number | 1 | 5 |
| | % | 12.5% | 5.4% |
| POC with Arias Stella reaction | number | 0 | 3 |
| | % | 0% | 3.2% |

The Fisher exact test statistic value is 0.35. The result is not significant at $p < 0.05$.

The residency of patients showed that (78%) of aborted women live in urban areas such as Taif, Jeddah, and Makkah, while (22%) of aborted women live in rural areas such as Trubah, Oshirah, Om Aldom, and Qia.

The prevalence of seropositivity was 5/22 (22%) in rural areas, while it was 3/78 (3.8%) in urban areas, as shown in Figure 1 (OR 5.9 (95% CI: 1.3; 26.6) $p < 0.05$).

Regarding the occupation of aborted women, (77%) of cases were unemployed, while the percentage of employed women was (8%). A total of (12%) of aborted women were teachers, (2%) of women were managers, and only (1%) was a student, as shown in Figure 2.

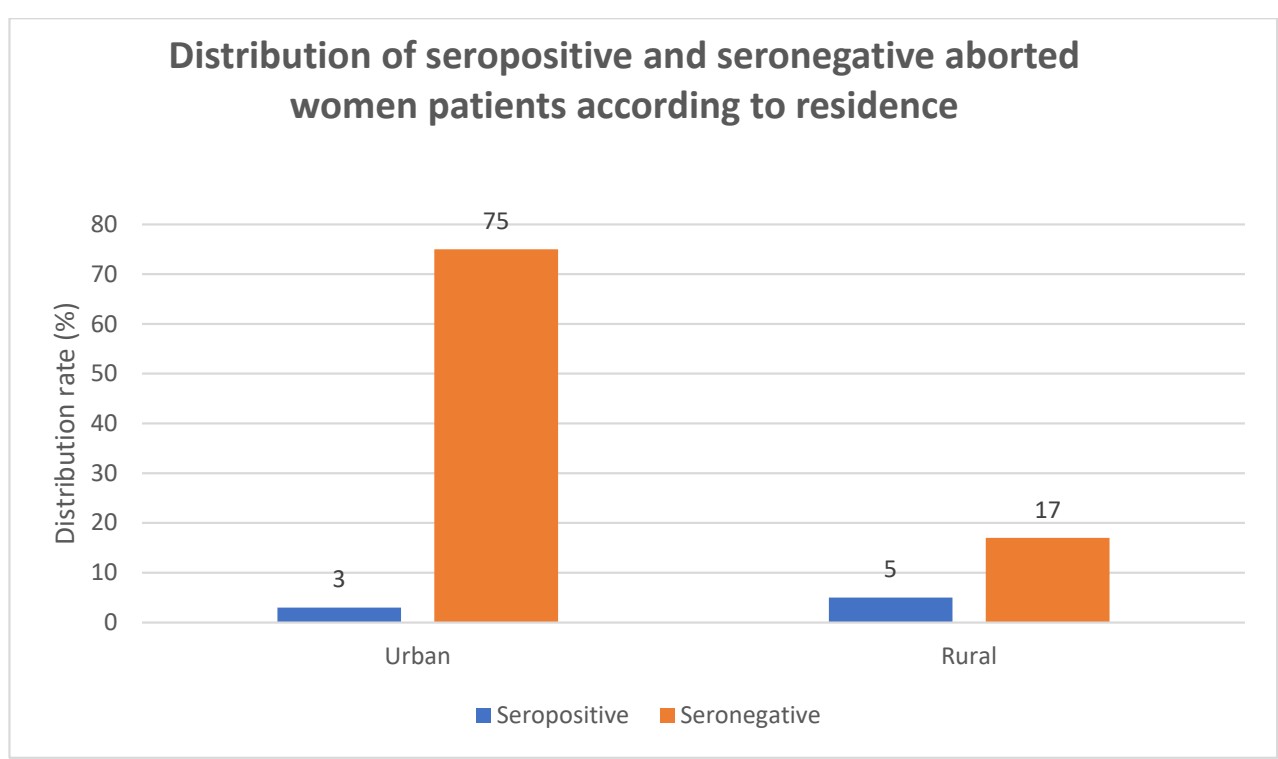

**Figure 1.** Distribution of seropositive and seronegative aborted women according to residence.

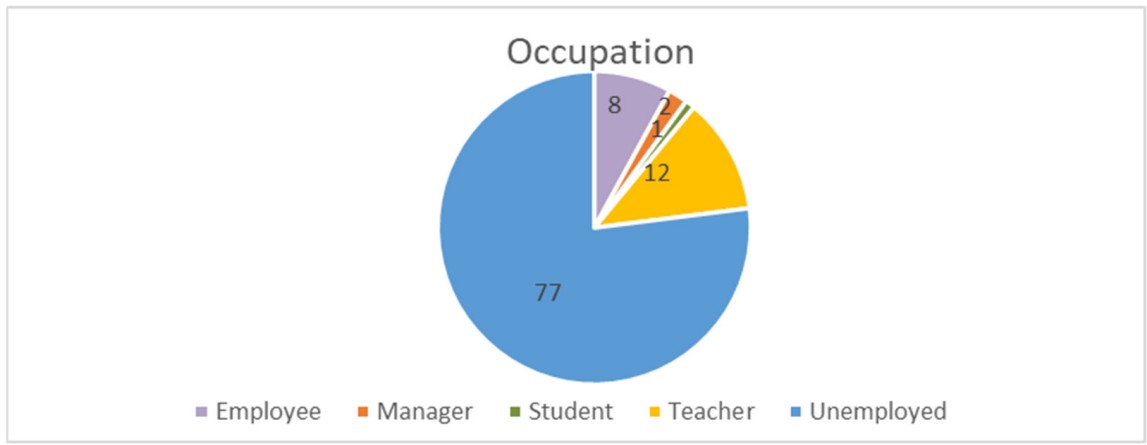

**Figure 2.** Distribution of aborted women according to occupation.

Student t-tests showed that there is a statistically significant increase in the mean count of white blood cells in seropositive women, in comparison with the mean count of white blood cells in seronegative women, as shown in Table 2.

**Table 2.** Hematological and chemical parameters of seropositive (+ve) and seronegative (−ve) aborted women.

|  | Serology | Mean | Std. Deviation | Std. Error Mean | *t*-Test (*p* Value) |
|---|---|---|---|---|---|
| WBC | +ve | 10.0350 | 4.14553 | 1.46567 | 0.034 * |
|  | −ve | 8.1151 | 2.75688 | 0.29223 |  |
| Urea | +ve | 18.0375 | 4.50807 | 1.59384 | 0.867 |
|  | −ve | 18.4135 | 6.16506 | 0.65350 |  |
| Creatinine | +ve | 0.3875 | 0.17219 | 0.06088 | 0.688 |
|  | −ve | 0.3565 | 0.21152 | 0.02205 |  |
| Sodium | +ve | 139.38 | 3.543 | 1.253 | 0.502 |
|  | −ve | 132.54 | 28.522 | 2.974 |  |

* Significant increase in the mean value of white blood cells in seropositive women.

## 4. Discussion

Miscarriage is described as a spontaneous abortion that occurs without the use of medical or mechanical means to end a pregnancy before the fetus has matured sufficiently to survive. Miscarriage, then, is a termination of a pregnancy before the 20th week of gestation. Additionally, a lot of pregnancies are lost before a woman ever recognizes she's pregnant [12].

A collection of pathogens known as TORCH includes *Herpes simplex virus*, *Cytomegalovirus*, *Toxoplasma*, and *Rubella*. These families of organisms infect pregnant women, which can result in preterm birth, intrauterine growth restriction, spontaneous abortion, and severe congenital defects with syndromic offspring in varying degrees. Because the developing fetus' immune system is unable to fight off the infectious organism, maternal infections, particularly those that occur early in pregnancy, can cause fetal loss or abnormalities. Numerous investigations have revealed a strong connection between maternal TORCH infections and pregnancy loss [13]. Pregnant women experience a decline in immunity due to changes in their endocrine systems, particularly a weakening of T lymphocyte immunological function, which makes them more vulnerable to contracting TORCH or having the virus reactivate in the future. The majority of them had negative fetal outcomes but only modest maternal morbidity [14]. The prevalence of these infections among pregnant subjects varies from one geographic region to another [15].

During pregnancy, the placenta limits vertical transfer and develops strong antimicrobial defense mechanisms. However, congenital disease-causing microbes have probably developed a variety of strategies to get around these defenses [16]. Early-pregnancy TORCH infections may cause congenital abnormalities, intrauterine growth restriction (IUGR), or fetal mortality [17]. Latent (asymptomatic) infections in newborns brought on by the later stages of pregnancy may subsequently manifest as symptoms of infection [18]. It is generally known that pregnancy causes a number of hormonal and immunological changes that raise a person's susceptibility to and severity of infections [17].

Our study included 100 aborted women that were admitted to King Faisal Medical Complex maternity hospital between the period 2018 and 2020. The percentage of women who had a miscarriage at less than 12 weeks of gestation (early miscarriage) was 65%, which is higher than that of women who miscarried between 12–20 weeks of gestation (late miscarriage), which was 35%, and the total seropositivity rate of TORCH infections and *hepatitis B* and C was 8% in Taif City. Among these organisms, the most common was *Hepatitis B virus* infections, whose seropositivity is 3%; further, total mixed TORCH infections were 3%. The current study shows the prevalence of mixed TORCH infections was higher in the first trimester of pregnancy (less than 12 weeks) in aborted women compared to abortion in the second trimester. This is due to the fact that, during this period, all the major organs and body systems are forming and can be damaged if the fetus is exposed to infectious agents.

The current results agreed with the study performed in Tehran, Iran, from 2012 to 2013, by Rasti et al. [17], which showed the majority of abortions occurred in the first trimester of pregnancy (49.4%, 40/81), while 44.4% (36/81) occurred in the second trimester. It was also discovered that women who had abortions had significantly higher *T. gondii* IgM seropositivity (3.6%). Compared with our study, *T. gondii* was 1% in mixed TORCH infection, which is higher than ours; this may be associated with eating raw/half-cooked meat and raw vegetable use.

In another study performed in the Hail region by Abd El-Galil, Metwally, and Al Shammary [19], the seroprevalence of TORCH among pregnant women in the study was 13%, and this is higher than our study and showed rising seropositivity to *Toxoplasma* in women with a bad obstetric history, which was caused by TORCH. This study disagrees with our study. This may be regarding the difference in geographical area and time of the study, the number of included cases, and lifestyles.

A study performed at Nobel Medical College Teaching Hospital by Lamichhane et al. [13] revealed that 41.74% were negative and 58.25% were positive for TORCH infections. In contrast to this study, the herpes virus was found to be among the most common causes of first-trimester spontaneous miscarriage in the eastern region of Nepal. The reason for this discrepancy may be related to the local geography and demographic differences.

A previous study in China, performed by Wang et al. [14], reported that 102 of the 1683 participants had a TORCH infection, with a total infection rate of 6.06% (102/1683), which is lower than our finding. They also found that CMV infection was more common than other pathogens, which is contrary to our study. There may be a connection between this and regional variations, economic and cultural circumstances, health situations, demographic differences, dietary practices, and lifestyles.

In this study, it was found that there was an increase in the percentage of women who had an abortion in urban areas, which was 78%, while the percentage of females who had an abortion in rural areas was 22%.

A previous study conducted in Tibet by Dang, Yan, and Zeng [20] agreed with our study. A total of 10,245 pregnancies involving a total of 3741 women were investigated. With an incidence rate of 3.9%, 386 spontaneous abortions were observed. Compared to women living in rural regions, women in urban areas had a greater rate.

Another study performed in China by Zhang et al. [21] showed that residency results are opposite to our results, wherein the study used multistep logistic regression and descriptive analysis to enroll 84,531 women from ten regions of China. In this study, rural areas had a higher probability of spontaneous abortion than cities did. Rural areas have 1.68 times higher SA risk than urban locations. The results may differ from our results due to environmental factors, lifestyle, and the fact that most of our cases were from urban areas.

This study, as shown in Figure 1, showed the prevalence rate of TORCH infection in rural areas was 22%, while it was 3.8% in urban areas, so the prevalence of TORCH infection in rural areas was higher than in urban areas.

In a study performed in Croatia by Vilibic-Cavlek et al. [22], *T. gondii*, CMV, and HSV-1 seroprevalence rates were considerably higher in rural women than in urban women (*T. gondii*: 44.0% vs. 25.4%, *p* < 0.001; CMV: 85.0% vs. 73.1%, *p* = 0.018; HSV-1: 86.0% vs. 76.4%, *p* = 0.041). This may be explained by the health services in rural areas, which are primitive.

In this study, Figure 2 showed that the percentage of unemployed women was (77%), which is a very high percentage compared to employed women (23%); 12% were teachers, 8% were employees, 2% were managers, and only 1% were students, which means that the occupation does not pose a threat to pregnancy and is not considered a compelling reason for abortion.

In contrast, a study previously conducted by Lemasters and Pinny [23] showed that, compared to unemployed pregnancies (11.7%), employed pregnancies had a significantly higher rate of spontaneous abortion (14.5%). (RR = 1.23, 95% CI = 1.02, 1.49).

In another study conducted on American nurses by Lawson et al. [24], the risk of spontaneous abortion and reported occupational exposures were examined; it reported 775 (10%) spontaneous abortions (within 20 weeks) and 6707 live births. Anti-neoplastic drug exposure was linked to a 2-fold increased risk of spontaneous abortion, particularly with early spontaneous abortion before the 12th week, and a 3.5-fold increased risk among nulliparous women after being adjusted for age, parity, shift work, and hours worked. In contrast to early spontaneous abortion, exposure to sterilizing drugs was associated with a 2-fold increased incidence of late spontaneous abortion (12–20 weeks). This difference between the results of our study and the two mentioned studies may be due to the nature of the work and the work environment.

Products of conception (POC) passed spontaneously or evacuated surgically or medically are usually submitted to histopathological investigation, which entails a gross and microscopic examination of tissues received in the histopathology laboratory [25]. The predominant histopathological diagnosis was normal POC, which constituted 59 (64%) in seronegative women. Miscarriage is a typical ailment, and, like many disorders, the right care depends on accurate diagnosis. The uterine products are subjected to a histopathologic examination as diagnostic tools to ascertain the type of miscarriage and to distinguish miscarriages from other disorders. If fetal tissues, trophoblasts, or chorionic villi were seen in addition to other tissues such as deciduae or secretory endometrium, an intrauterine pregnancy was considered confirmed [26].

Histopathological diagnosis, as shown in Table 1 and Figure 3, shows normal POC in seronegative aborted women was 59 (64.1%) and constituted the highest percentage, followed by degenerated POC, which was 12 (13%) in seronegative, and then POC with mixed inflammatory infiltrates and chronic endometritis, which was seven (7.6%) in seronegative; septic abortion was six (6.5%) in seronegative, POC with hydropic changes was five (5.4%) in seronegative, POC with mixed inflammatory infiltrates and chronic endometritis was four (50%) in seropositive, POC with Arias Stella reaction was three (3.2%) in seronegative, degenerated POC was two (25%) in seropositive, and septic abortion and POC with hydropic changes in seropositive was one (12.5%).

Normal POC was formed from sheets of decidua and chorionic villi (1). Septic POC shows decidual sheets with dense neutrophilic acute inflammatory infiltrates and dilated congested blood vessels (2). POC with chronic endometritis shows that decidua and endometrial stroma are densely infiltrated by mixed chronic inflammation, which infiltrates and fibrosis blood vessels (3). Degenerated retained POC shows degenerated chorionic villi and decidua with inflammatory infiltrates (4). POC with hydropic changes shows enlarged chorionic villi with pale edematous cores (5). POC with an Arias Stella reaction shows sheets of decidua and dilated secretory endometrial glands lined by hypertrophied vacuolated epithelial cells with hyperchromatic nuclei (6).

A study in Arar conducted by Hassan, Hegazy, and Mosaed [27] revealed the decidual reaction represents about 50 cases (25%), which is less than our results in normal POC, which was 59 (64.1%) in seronegative aborted women. In 10 cases (5%), there is hydropic alteration due to villous vascular supply loss, which disagrees with our study. POC with hydropic changes was five (5.4%) in seronegative, and one (12.5%) in seropositive cases, and septic abortion was observed 17 cases (8.5%); in contrast to our study, seronegative aborted women with septic abortion were six (6.5%), and seropositive was one (12.5%).

Another study in Libya, performed by Ashour, Gheryani, and Meidan [28], reported that 19% of chorionic villi showed hydropic changes, which was higher than our study, in which the hydropic changes in seropositive was one (12.5%), and in seronegative, it was five (5.4%).

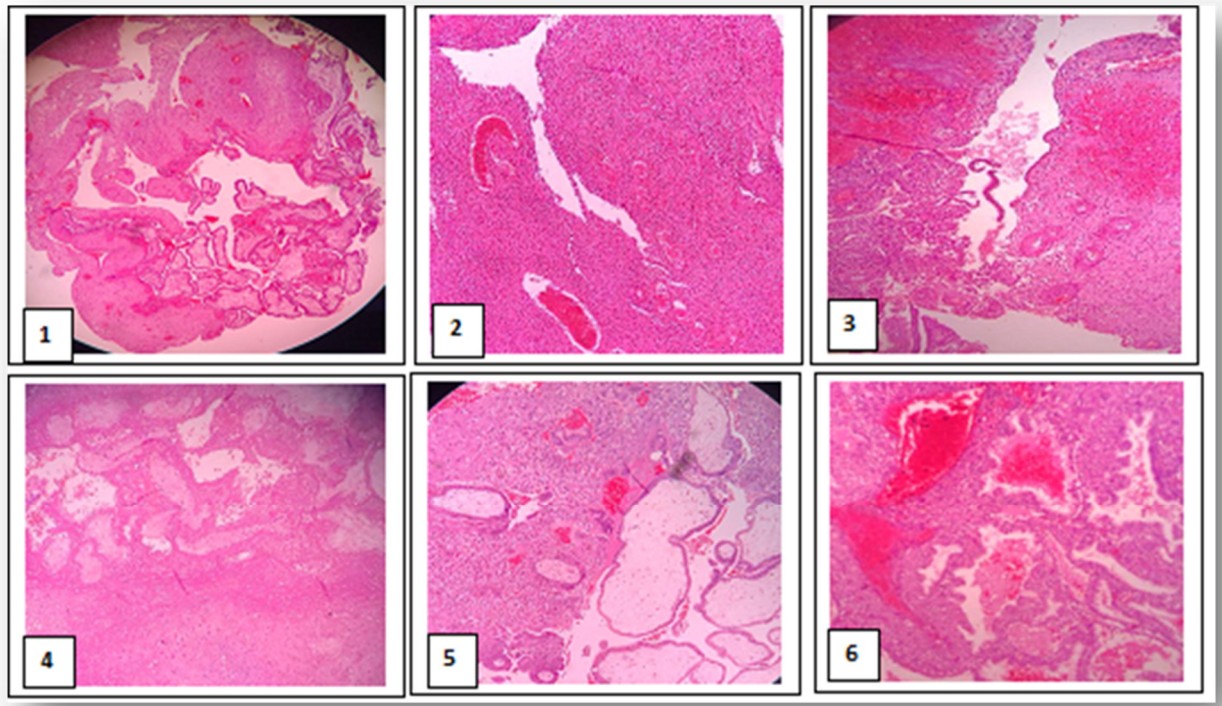

**Figure 3.** (**1**–**6**) Microscopic appearance of POC from aborted women (Photomicrograph of 4 microns thick H&E-stained paraffin section 100×).

Compared to a study performed in Nigeria conducted by Hayi and Onyishi [25] reported normal POC constituted the most common histopathological diagnosis at 73 (68.2%); these results disagree with our results in normal POC, which was 59 (64.1%) in seronegative aborted women. Arias Stella reaction was eight (7.5%), which is higher than our results in seronegative aborted women with POC with Arias Stella reaction, which was three (3.2%); variation may be due to the geographical area and population differences.

In order to rule out pathological issues such as anemia, thrombocytopenia, bleeding disorder, thrombosis, and thrombophilia, a complete blood count (CBC) test is frequently used and advised during the early stages of pregnancy. Important CBC values include the kinds of white blood cells, which alter during pregnancy with a considerable increase in the ratio of granulocytes to T helper (Th)-1 lymphocytes and a concurrent decrease in the ratio of Th-2 lymphocytes to monocytes [29].

A frequent and dangerous consequence of pregnancy is renal dysfunction. Numerous coordinated changes that influence the renal structure and hemodynamics are part of a healthy pregnancy. There is still much to learn about these alterations; therefore, more research is needed in this area. Renal failure during pregnancy can be caused by a variety of illnesses, posing problems for both the mother and the unborn child. Prerenal dysfunction can result from blood loss from pregnancy-related problems such as prenatal hemorrhage or fluid losses, leading to severe vomiting, as in hyperemesis gravidarum [30].

In our study, we collected data from one hundred aborted women. This increaselogy lab. CBC reports in Table 2 showed a significant increase in the mean count of white blood cells (WBC) in seropositive women mean ± stander deviation (SD) was 10.03 ± 4.14, in comparison with the mean count of white blood cells in seronegative women mean ± stander deviation (SD) was 8.11 ± 2.75. These increases were associated with WBC due to defending the body against infections and disease, and it usually means there is an infection or inflammation in the body, as shown in seropositive women.

The current results agreed with the study provided by Al-Husban et al. [31], which occurred between June 2017 and September 2018 at the Jordan University Hospital, which demonstrates that an inflammatory environment can be linked to an increase in WBC in

peripheral blood, which can result in preeclampsia and poor pregnancy outcomes. Additionally, it was discovered that women with ovarian hyper-stimulation syndrome who had raised WBC levels exceeding $15 \times 10^6$ L in their peripheral blood had a higher chance of miscarriage. The current study demonstrates that there is no statistically significant difference between urea, creatinine, and sodium in both seropositive and seronegative aborted women. Elevated WBC counts in the first trimester are associated with an increased risk of missed miscarriage, and this can serve as an early warning for adverse pregnancy outcomes.

## 5. Conclusions

This study concluded that the first trimester of pregnancy was when miscarriages were most prevalent, the prevalence of TORCH infections was 8% in aborted women in Taif City, and *Hepatitis B virus* (HBV) and total mixed TORHC infection constituted the highest percentage of TORCH pathogens in aborted women. The histopathology was an effective indicator for infectious causes of abortion as a majority of seropositive cases showed mixed inflammatory infiltrates in their POC. The prevalence of TORCH infections was higher in aborted women living in rural areas than in aborted women living in urban areas, and contact with pet animals was a risk factor for an infectious cause of abortion, so it is important to provide health campaigns to bring awareness to the population about the risk factors of infectious agents to be avoided, especially during pregnancy.

**Author Contributions:** Conceptualization, A.M.M. and K.A.I.; Formal analysis, A.S.A.-G. and N.S.A.; Investigation, A.M.A. and A.A.A.; Methodology, H.M.H., S.H.A.-a. and A.K.A.; Resources, N.F.J., H.H.A., A.A.E. and K.H.K.; Supervision, K.A.I.; Writing—original draft, A.M.M. and O.M.K.; Writing—review and editing, H.M.H. and R.A.R. All authors have read and agreed to the published version of the manuscript.

**Funding:** This research received no external funding.

**Institutional Review Board Statement:** The College of Applied Medical Sciences' research committee approved the study's execution—Taif University—after an explanation of the aim of the study. Further, the study was approved by the ethics committee of the King Faisal Medical Complex maternity hospital (H-02-T-123).

**Informed Consent Statement:** Not applicable.

**Data Availability Statement:** Data available on request from Khadiga Ahmed Ismail.

**Conflicts of Interest:** The authors declare no conflict of interest.

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
