# Peer review of "Prevalence of Infectious Agents Causing Abortion in Pregnant Women Using Serological Tests and Histopathological Analysis"

_2673-8007, doi:10.3390/applmicrobiol3030048_

Round 1

Reviewer 1 Report

Thanks to the authors for taking up this very important topic in this manuscript 'Prevalence Of Infectious Agents Causing Abortion In Pregnant Women Using Serological tests and Histopathological analysis'.

The manuscript is very clear, easy undertsandable fomulated keeping to a very clear structure.

In the introduction, the abstract is fundameted by references and deeper classification. Very well formulated background leading to the topic of the study.

Yet I have few comments:

(1) In your introduction you state the importance of histopathology.

'Histopathological examination is essential to identify previously unsuspected diseases first-trimester specimens differ greatly in their composition. Histopathology is an integral and routine component of the management of patients with sporadic and recurrent early pregnancy failure [11].'

Could you please addit some words about the strength and limitations of histopathology in cases of infection correlating with abortion?

(2): could you please specify the material which was sent to pathology examniation: gestational sac? decidua? mixed? Did the pathologists follow a standardized procedure?

(3): Fig.3 shows the difference between seroneagtive to seropositive, but the legend in the diagram assigns the colours to 'infected' and 'uninfected'.

You should consider to align.

(4): The histological photos seem to be out of focus...

Overall, the discussion consists mainly of comparing other study results to this study.

I would appreciate, if the comparisons couild be shortened and filled up with more own ideas, reflections, interpretation of the findings, differences and accordance.

(5): In the discussion the authors mention that histopathological examination is important '... to differentiate miscarriages from other conditions'.

Which?

(6): next sentence: 'An intrauterine pregnancy was confirmed if fetal tissues, trophoblasts, or chorionic villi were identified...' 

chorionic villi with/without fetal vessels?

(7): in the following paragraph the authors are summing up the different histological findings.

Recommendation to the authors: more focus on discussion. 

(8): Next, the authors state 'Arias-Stella reaction was 8(7.5%) which is higher than our results in seronegative aborted women with POC with Arias-Stella reaction were 3 (3.2%), variation may be due to geographical area and population differences '. 

I can't believe in this. You should consider differences of the material and sampling made by the pathologist.

(9): Conclusion: the authors state that 'The histopathology was an effective indicator for infectious causes of abortion as a majority of seropositive cases showed mixed inflammatory infiltrates in their POC'.

To state this, it is important to have direct correlation to the infectious agens or specific immuno cell response. Can you support this by immunostain?

Otherwise it may reflect unspecific inflammation? Can you exclude host-versus-graft reaction as a cause of abortion?

Author Response

Response to reviewer 1 comments

Thanks to the authors for taking up this very important topic in this manuscript 'Prevalence Of Infectious Agents Causing Abortion In Pregnant Women Using Serological tests and Histopathological analysis'.

The manuscript is very clear, easy undertsandable fomulated keeping to a very clear structure.

In the introduction, the abstract is fundameted by references and deeper classification. Very well formulated background leading to the topic of the study.

Yet I have few comments:

(1) In your introduction you state the importance of histopathology.

'Histopathological examination is essential to identify previously unsuspected diseases first-trimester specimens differ greatly in their composition. Histopathology is an integral and routine component of the management of patients with sporadic and recurrent early pregnancy failure [11].'

Could you please addit some words about the strength and limitations of histopathology in cases of infection correlating with abortion?

A: Chronic inflammation detected in histopathology indicate presence of inflammation may be due to infectious causes confirmed by serology.

(2): could you please specify the material which was sent to pathology examniation: gestational sac? decidua? mixed? Did the pathologists follow a standardized procedure?

A: We follow a standardized procedure. Most cases sent to histopathology examination macroscopically have multiple irregular dark brownish soft tissue fragments with no fetal parts or gestational sacs detected .Some cases sent as spongy dark brownish tissue fragments

(3): Fig.3 shows the difference between seroneagtive to seropositive, but the legend in the diagram assigns the colours to 'infected' and 'uninfected'.

You should consider to align.

A: considered and corrected

(4): The histological photos seem to be out of focus...

Overall, the discussion consists mainly of comparing other study results to this study.

I would appreciate, if the comparisons couild be shortened and filled up with more own ideas, reflections, interpretation of the findings, differences and accordance.

A: will be considered

(5): In the discussion the authors mention that histopathological examination is important '... to differentiate miscarriages from other conditions'.

Which?

A: Hormonal disturbance and extrauterine tubal pregnancy

(6): next sentence: 'An intrauterine pregnancy was confirmed if fetal tissues, trophoblasts, or chorionic villi were identified...'

chorionic villi with/without fetal vessels?

A: yes

(7): in the following paragraph the authors are summing up the different histological findings.

Recommendation to the authors: more focus on discussion.

A: will be considered

(8): Next, the authors state 'Arias-Stella reaction was 8(7.5%) which is higher than our results in seronegative aborted women with POC with Arias-Stella reaction were 3 (3.2%), variation may be due to geographical area and population differences '.

I can't believe in this. You should consider differences of the material and sampling made by the pathologist.

A: will be considered

(9): Conclusion: the authors state that 'The histopathology was an effective indicator for infectious causes of abortion as a majority of seropositive cases showed mixed inflammatory infiltrates in their POC'.

To state this, it is important to have direct correlation to the infectious agens or specific immuno cell response. Can you support this by immunostain?

A: Further study will be considered for this correlation

Otherwise it may reflect unspecific inflammation? Can you exclude host-versus-graft reaction as a cause of abortion?

Reviewer 2 Report

The authors present an interesting study evaluating the incidence of TORCH infections in cases of miscarriage. The main limitation of the study is the limited number of included patients and seroconversions. The authors included only 8 cases of seroconversion in 100 included patients. Unless Authors can increase the cohort by adding at least 400-500 cases of miscarriage, I suggest simplifying the analyses which appear to be redundant. In particular, please remove the stratification by early and late miscarriage and the distribution by age which is useless since most of the boxes are with value 0.

Abstract: Correct TORHC with TORCH. Clarify POC in the abstract and the first time you mention it in the text.

Introduction: Toxoplasma infection is associated not only with fetal death and miscarriage, but also with congenital malformations with a transmission rate of about 30% (PMID: 31404785, PMID: 34458947).

Data analysis: please write a paragraph stating which associations or comparison authors have made.

Results: “Our study included a total of 100 aborted women. The percentage”

“The prevalence of seropositivity was 5/22 (22%) in rural areas while it was 3/78 (3.8%)”: add the Odds ratio and relative p value to demonstrate whether there is a significant difference in seropositivity prevalance according to the urban/rural setting.

From “Normal POC formed.. to … hyperchromatic nuclei” should belong to the methods, where you describe the histopathological features of tissue from miscarriages. In the Results section please describe the findings from Table 3, with Odds Ratio analysis and p values. It seems that miscarriages with seropositivity show more abnormal istopathological features compared to controls: this would add value to the manuscript. If the statistical analysis confirm this impression, this concept should be highlighted also in the Discussion section.

Table 4 is missing, please correct Table 5 in the case. In table 5 it is not clear the second p value (eg 0.237 for WBC). There should be only one p value for t Student’s test (“t-test”is mispelled).

Discussion: in this section Authors should not include paragraphs explaining on the Results of the study, which belong to the Results section, but only discuss these  results and compare them with the current literatures. Therefore, please move from this section paragraphs at Page 11 from “Histopathological diagnosis… to… seropositive was 1(12.5%)” as mentioned before, and just Discuss the importance of these results.

Moderate English language check is required

Author Response

Response to reviewer 2 comments

The authors present an interesting study evaluating the incidence of TORCH infections in cases of miscarriage. The main limitation of the study is the limited number of included patients and seroconversions. The authors included only 8 cases of seroconversion in 100 included patients. Unless Authors can increase the cohort by adding at least 400-500 cases of miscarriage, I suggest simplifying the analyses which appear to be redundant. In particular, please remove the stratification by early and late miscarriage and the distribution by age which is useless since most of the boxes are with value 0.

A: Done

Abstract: Correct TORHC with TORCH. Clarify POC in the abstract and the first time you mention it in the text.

A: Done

Introduction: Toxoplasma infection is associated not only with fetal death and miscarriage, but also with congenital malformations with a transmission rate of about 30% (PMID: 31404785, PMID: 34458947).

Data analysis: please write a paragraph stating which associations or comparison authors have made.

A: Done

Results: “Our study included a total of 100 aborted women. The percentage”

“The prevalence of seropositivity was 5/22 (22%) in rural areas while it was 3/78 (3.8%)”: add the Odds ratio and relative p value to demonstrate whether there is a significant difference in seropositivity prevalance according to the urban/rural setting.

A: OR 5.9(95%CI:1.3;26.6) p<0.05

From “Normal POC formed.. to … hyperchromatic nuclei” should belong to the methods, where you describe the histopathological features of tissue from miscarriages. In the Results section please describe the findings from Table 3, with Odds Ratio analysis and p values. It seems that miscarriages with seropositivity show more abnormal istopathological features compared to controls: this would add value to the manuscript. If the statistical analysis confirm this impression, this concept should be highlighted also in the Discussion section.

Table 4 is missing, please correct Table 5 in the case. In table 5 it is not clear the second p value (eg 0.237 for WBC). There should be only one p value for t Student’s test (“t-test”is mispelled).

A: corrected

Discussion: in this section Authors should not include paragraphs explaining on the Results of the study, which belong to the Results section, but only discuss these  results and compare them with the current literatures. Therefore, please move from this section paragraphs at Page 11 from “Histopathological diagnosis… to… seropositive was 1(12.5%)” as mentioned before, and just Discuss the importance of these results.

A: Done

Round 2

Reviewer 2 Report

It is very confusing because Authors state they made some of the requested changes but I did not find them in the revised manuscript. Please revise carefully the manuscript again and in the revision letter state where your changes were done (Page, Lines).

-Introduction: Toxoplasma infection is associated not only with fetal death and miscarriage, but also with congenital malformations with a transmission rate of about 30% (PMID: 31404785, PMID: 34458947).

Authors state they mentioned this concept but I didnt find it in the text, please revise or highlight were changes were made.

-Data analysis: please write a paragraph stating which associations or comparison authors have made.

Another point not really addressed. Authors need to clarify which kind of associations or comparison they have done in the Methods. Readers need to understand from the Methods which kind of associations they are going to find in the Results (eg. We aimed to compare the incidence of miscarriage in the population, then we compared the incidence of miscarriange between A and B, and we used this stastistical analysis to perform this comparison etc etc etc). You cannot simply state “statistical analyses were made”, it is quite obvious in a scientific paper that you made some statistical analysis.

-Results: “Our study included a total of 100 aborted women. The percentage” You need a point after women, then  start a new sentence with The percentage.

-From “Normal POC formed.. to … hyperchromatic nuclei” should belong to the methods, where you describe the histopathological features of tissue from miscarriages. In the Results section please describe the findings from Table 3, with Odds Ratio analysis and p values. It seems that miscarriages with seropositivity show more abnormal istopathological features compared to controls: this would add value to the manuscript. If the statistical analysis confirm this impression, this concept should be highlighted also in the Discussion section.

This point was  not addressed

Minor English editing required

Author Response

It is very confusing because Authors state they made some of the requested changes but I did not find them in the revised manuscript. Please revise carefully the manuscript again and in the revision letter state where your changes were done (Page, Lines).

-Introduction: Toxoplasma infection is associated not only with fetal death and miscarriage, but also with congenital malformations with a transmission rate of about 30% (PMID: 31404785, PMID: 34458947).

A: I mentioned a paragraph about TORCH incidence that starts with: According to a study by Al-Hakami et al.

Authors state they mentioned this concept but I didnt find it in the text, please revise or highlight were changes were made.

-Data analysis: please write a paragraph stating which associations or comparison authors have made.

A: I highlighted it in Data analysis section

Another point not really addressed. Authors need to clarify which kind of associations or comparison they have done in the Methods. Readers need to understand from the Methods which kind of associations they are going to find in the Results (eg. We aimed to compare the incidence of miscarriage in the population, then we compared the incidence of miscarriange between A and B, and we used this stastistical analysis to perform this comparison etc etc etc). You cannot simply state “statistical analyses were made”, it is quite obvious in a scientific paper that you made some statistical analysis.

A: All included cases were aborted women and there weren’t any women with normal pregnancy outcome to make correlation (no control)

-Results: “Our study included a total of 100 aborted women. The percentage” You need a point after women, then  start a new sentence with The percentage.

A: Highlighted

-From “Normal POC formed.. to … hyperchromatic nuclei” should belong to the methods, where you describe the histopathological features of tissue from miscarriages. In the Results section please describe the findings from Table 3, with Odds Ratio analysis and p values. It seems that miscarriages with seropositivity show more abnormal istopathological features compared to controls: this would add value to the manuscript. If the statistical analysis confirm this impression, this concept should be highlighted also in the Discussion section.

This point was  not addressed

A: There is no table 3 in the revised manuscript